# FLD+: Data-efficient Evaluation Metric for Generative Models

## Abstract

We introduce a new metric to assess the quality of generated images that is more reliable, data-efficient, compute-efficient, and adaptable to new domains than the previous metrics, such as Fréchet Inception Distance (FID). The proposed metric is based on normalizing flows, which allows for the computation of density (exact log-likelihood) of images from any domain. Thus, unlike FID, the proposed Flow-based Likelihood Distance Plus (FLD+) metric exhibits strongly monotonic behavior with respect to different types of image degradations, including noise, occlusion, diffusion steps, and generative model size. Additionally, because normalizing flow can be trained stably and efficiently, FLD+ achieves stable results with two orders of magnitude fewer images than FID (which requires more images to reliably compute Fréchet distance between features of large samples of real and generated images). We made FLD+ computationally even more efficient by applying normalizing flows to features extracted in a lower-dimensional latent space instead of using a pre-trained network. We also show that FLD+ can easily be retrained on new domains, such as medical images, unlike the networks behind previous metrics – such as InceptionNetV3 pre-trained on ImageNet.

## 1 Introduction

**Numerous image generation models have emerged** for applications, such as inpainting, and generating images from text prompts, text-guided image manipulation etc., and has been capturing both industry interest and public imagination in recent years. With the rise of these large-scale image generation models, assessing their quality has become essential for meaningful comparisons. To effectively compare individual generated images and assess the overall performance of these models, a reliable evaluation metric is vital for measuring how closely generated images match the real ones. This is essential to determine if generative models can create images that look authentic to human viewers – a critical objective across many applications. Contemporary generative models, which predominantly rely on diffusion or adversarial training, cannot be effectively or efficiently evaluated using existing metrics (Jayasumana et al., 2023).

**Assessing realism of generated images is particularly hard** compared to other computer vision tasks. Generated images must be judged on several aspects — quality (e.g., sharpness), micro-artifacts (e.g., realistic texture and edges), macro-artifacts (e.g., realistic object parts, proportions, and spatial relationships), and diversity (e.g., avoiding mode collapse). These aspects are inherently hard to evaluate objectively as these are rooted in human perception. Furthermore, human evaluations are costly, labor-intensive, and impractical to scale for large datasets. As a result, researchers increasingly rely on automated evaluation techniques that offer a more resource-efficient way to measure and compare model performance, providing a quantifiable gauge of quality. However, due to the abovementioned challenges, there is no universally accepted metric to assess the quality of generated images.

**An ideal metric should possess certain key attributes** for evaluating generative models. Firstly, it should increase consistently as noise or image degradation intensifies. Moreover, given that modern generative models produce images with such subtle imperfections, which even human observers struggle to detect, the metric must be sensitive to even slightest of distortions. Furthermore, it should provide

reliable assessments with a limited sample of generated images, thus allowing the average metric value to stabilize quickly under consistent conditions. Finally, the metric should be computationally efficient, making it suitable for integration into validation steps during training iterations.

**Despite their limitations, distribution-distance based metrics remain popular**, especially Fréchet Inception Distance (FID) (Heusel et al., 2017), for automatically evaluating generative models. FID calculates the distance between the feature distributions of real and generated images by approximating each as a multivariate Gaussian and computing the Fréchet Distance (FD) between these distributions. It relies on activations from the penultimate layer of an InceptionV3 network pre-trained on ImageNet, which allows comparisons at an object or semantic level. However, FID overlooks artifacts in fine-grained image details (Jayasumana et al., 2023). Experiments on text-to-image models have shown that FID also often disagrees with human ratings and displays non-monotonic behavior with increasing degradation, limiting its effectiveness in evaluating generated image quality (Jayasumana et al., 2023). Furthermore, statistical tests and empirical studies highlight FID's inability to fully capture the subtle nuances of image generation (Jayasumana et al., 2023). FID can also exhibit bias based on the model being evaluated. Moreover, it requires a large sample size (over 20,000) of generated and real images to yield stable and reliable metric values, posing challenges in scenarios with limited samples or when computational efficiency is necessary, such as during validation steps within training iterations. Despite its known limitations, FID's simplicity and ease of use have made it a standard metric in the field.

**Recently, normalizing flows have been employed** to learn the real data distribution and estimate the likelihood of generated data, thus providing a metric, such as Flow-based Likelihood Distance (FLD), for quantifying the distance between real and generated data distributions (Jeevan et al., 2024). However, as normalizing flows operate in large latent spaces with dimensions equal to the input, processing images through these models is computationally intensive.

**Our proposed metric, FLD+**, leverages a pre-trained backbone network to extract a feature tensor that captures essential information from images. Only this lower-dimensional feature tensor is then passed through a normalizing flow which increases the speed of training and evaluation. Thus, FLD+ evaluates the alignment between the distributions of generated and real images, offering a more accurate and efficient measure of their similarity.

**The main contributions** of this paper are:

- Propose FLD+, a new metric for evaluating generated image based on normalizing flows.

- Hypothesize and confirm the sensitivity and monotonicity of FLD+ to a diverse set of image distortions, diffusion steps, and generative model sizes.

- Hypothesize and confirm data efficiency of FLD+ (by two orders of magnitude compared to FID) for stable metric estimation due to the image log-likelihood estimation by normalizing flows.

- Use normalizing flow to model the distribution of feature space using a pre-trained feature extractor for enhanced training efficiency.

- Hypothesize and confirm the ability to adapt to new image domains with small data and compute requirements even while using a pre-trained feature extractor. [1]

## 2 Previous Metrics and their Limitations

Numerous evaluation metrics have been developed to assess generative models, including inception score (IS) (Salimans et al., 2016), kernel inception distance (KID) (Bińkowski et al., 2018), FID (Heusel et al., 2017), perceptual path length (Karras et al., 2021), Gaussian Parzen window (Goodfellow et al., 2014), clip maximum mean discrepancy (CMMD) (Jayasumana et al., 2023) as well as human annotation techniques, such as HYPE (Zhou et al., 2019). Among these, IS and FID have seen widespread adoption. The Inception

---

[1]Our code is available at `XXXX`

Score leverages an InceptionV3 model trained on ImageNet-1k to evaluate both the diversity and quality of generated images by analyzing their class probabilities. One notable advantage of the Inception Score is that it can be computed without requiring real images for reference, making it useful in scenarios where real image data is unavailable.

## 2.1 Limitation 1: Normality Assumption

In calculating FID, the Fréchet Distance (FD) assumes that the InceptionV3 embeddings follow a multivariate normal distribution in order to apply a closed-form FD solution (Dowson & Landau, 1982). When image embeddings significantly deviate from this assumption, which is often the case in real scenarios, the resulting FID scores can become unreliable and biased. Additionally, estimating large covariance matrices (e.g., $2048 \times 2048$) from limited samples of real and generated images further amplifies potential errors, leading to *inconsistent evaluations.*

This issue becomes apparent when we compute the FD between an isotropic Gaussian and a mixture-of-Gaussians with matching overall means and covariance matrices (Jayasumana et al., 2023; Jeevan et al., 2024). The mixture distributions comprises a mixture of four Gaussians (which violates the normality assumption), each with the same mean and covariance as the isotropic Gaussian. While the FID between such pairs is 0, an ideal metric should be able to indicate that these are different distributions.Even the unbiased version of FID proposed in previous work (Chong & Forsyth, 2019) is subject to this limitation, as it also relies on the strong assumption of normality in embeddings. Consequently, these findings highlight the need for alternative metrics that can accommodate distributions of real images, unless the feature layers are trained to adhere to a Gaussian.

MMD and FLD has demonstrated advantages over FID by bypassing the assumption that distributions must be multivariate Gaussians.

## 2.2 Limitation 2: Large Data and Compute

Both KID (Bińkowski et al., 2018) and FID require real images or their large covariance matrices alongside generated images for evaluation, as they assess the similarity between the distributions of real and generated data. KID calculates the squared maximum mean discrepancy (MMD) to quantify this similarity, while FID measures the squared FD between the two distributions. These metrics try to calculate how closely the generated images resemble the real ones for evaluating the quality of generative models. Reliable estimates of the distribution of the generated images require a large amount of data for a stable estimate (Jayasumana et al., 2023).

## 2.3 Limitation 3: Reliance on ImageNet Training

IS, FID, and KID rely on embeddings from the InceptionV3 model, trained on the 1.3 million images and 1,000 classes in the ImageNet-1k dataset (Szegedy et al., 2015). This constraint restricts their ability to fully represent the broader and more complex range of images that modern generative models can produce. Consequently, these metrics are less effective for evaluating certain types of generated content, particularly in domains that extend beyond ImageNet's class and content diversity. Several studies have highlighted the limitations and unreliability of these evaluation metrics for image generation, with a particular focus on FID (Chong & Forsyth, 2019). Previous research has shown that FID can act as a biased estimator, with scores highly sensitive to low-level image processing operations, such as compression and resizing, which can significantly alter FID values (Parmar et al., 2022).

# 3 Proposed Metric using Normalizing Flows

Since estimating the likelihood of a generated image with respect to the distribution of real images seems like a logical foundation to structure a metric of image generation quality, we turn our attention to normalizing flows. We then describe how to structure a metric and improve its computational efficiency for both training and testing.

### 3.1 Background on Normalizing Flows

Normalizing flows are a class of generative models built on invertible transformations. Unlike other popular generative models, such as GANs, VAEs, and diffusion models, normalizing flows are unique in their ability to compute the exact likelihood of data efficiently (Prince, 2023; Kobyzev et al., 2021). Other models do not explicitly estimate the likelihood: VAEs only approximate a lower bound on the likelihood, while GANs provide a sampling mechanism without a likelihood estimate (Kobyzev et al., 2021).

Normalizing flows calculate the likelihood of a given sample $\mathbf{x}$ by transforming its complex data distribution into a simpler one—typically a Gaussian—through a series of invertible mappings. Each transformation repositions the data in a new space while retaining the ability to compute its log-likelihood $p_{\mathbf{x}}(\mathbf{x})$ in the original space, using the change of variables formula:

$$\log p_{\mathbf{x}}(\mathbf{x}) = \log p_{\mathbf{z}}(f(\mathbf{x})) + \log \left| \det \left( \frac{\partial f(\mathbf{x})}{\partial \mathbf{x}} \right) \right|,$$

where $f(\mathbf{x})$ denotes the transformation applied by the normalizing flow, $p_{\mathbf{z}}(f(\mathbf{x}))$ is the likelihood of the transformed sample in the latent space (such as the probability density under a Gaussian distribution), and $\det \left( \frac{\partial f(\mathbf{x})}{\partial \mathbf{x}} \right)$ is the Jacobian determinant, which adjusts for changes in volume during transformation (Kobyzev et al., 2021). This exact likelihood computation allows normalizing flows to model data distributions with high fidelity and flexibility. To learn a data distribution, we train the parameters of the normalizing flow $f(\mathbf{x})$ by maximizing the likelihood of the training data (Prince, 2023). This process optimizes the model to accurately represent the underlying data distribution.

These properties of normalizing flows allow us to both (1) obtain a stable average metric with far fewer test images than those needed by other methods, and (2) fine-tune a stand-alone network on new domains with just a few thousand images.

### 3.2 Flow-based Likelihood Distance Plus (FLD+)

To addresses the limitations of previous metrics for assessing the quality of generated images, we propose a metric called Flow-based Likelihood Distance Plus (FLD+). We improve upon Flow-based Likelihood Distance (FLD) (Jeevan et al., 2024), whose key drawback is that normalizing flows require the latent space to match the input's dimensionality, which can result in high computational costs and memory usage for large, high-dimensional inputs, such as images. This constraint limits the scalability and practical applicability of normalizing flow-based metrics for evaluating complex, high-resolution generative models.

The concept behind the FLD+ metric, where we measure the distance between real and generated distributions by calculating the likelihood of generated data with respect to the real distribution, is illustrated in Fig. 1. This approach leverages the likelihood as a direct indicator of how closely the generated data aligns with the real data, providing a more accurate and distribution-specific metric for evaluation.

We first take a set of real images and a set of generated images.

$$\mathcal{R} = \{r_1, r_2, \ldots, r_n\} \qquad \text{(set of real images)}$$

$$\mathcal{G} = \{g_1, g_2, \ldots, g_m\} \qquad \text{(set of generated images)}$$

We design a flow model $F$ that combines an ImageNet pre-trained computer vision backbone, $B$, as a feature extractor to reduce the dimension of the latent space to be modeled using a normalizing flow, $N$. The weights of $B$ are kept frozen, and features, $\mathbf{x}_b$ are extracted just before the final layer (Eq. 1). Following $B$, we add a 2D average pooling layer $p$ to down-sample the output feature tensor (Eq. 2), which is then flattened into a vector $\mathbf{x}_f$ (Eq. 3) and passed to the normalizing flow $N$. The normalizing flow $N$ is parameterized by a set of trainable parameters, $\theta$, allowing it to learn the data distribution based on the extracted features (Eq. 4) and output the likelihood of data based on the real distribution.

$$\mathbf{x}_b = B(\mathbf{x}_{in}) \qquad \mathbf{x}_{in} \in \mathcal{R}, \mathbf{x}_b \in \mathbb{R}^{H \times W \times C} \tag{1}$$

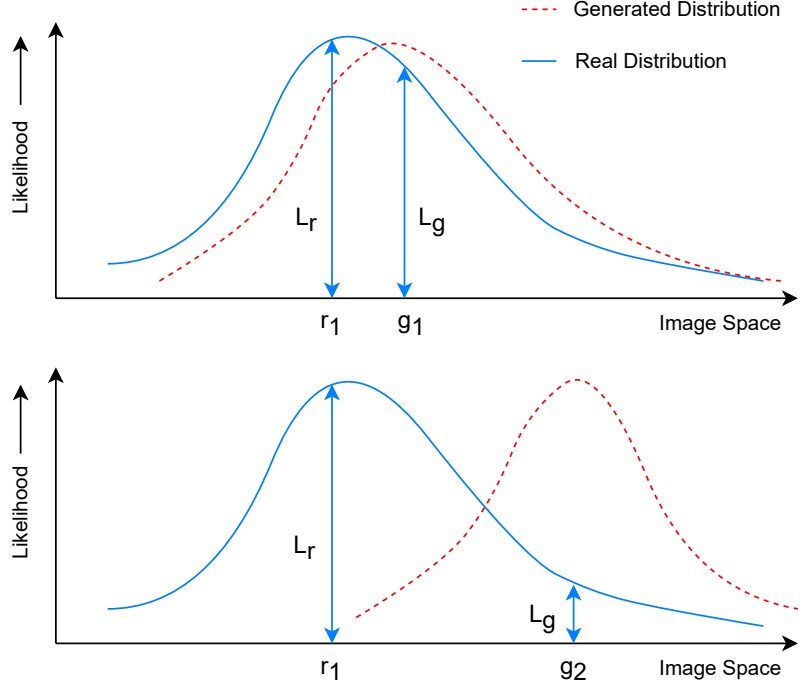

Figure 1: The visual interpretation of the FLD+ metric. The real data distribution (blue curve) is modeled by the flow model, and the generated image distribution (dotted red curve) is not modeled by it. For generated images, most data points will fall within high-likelihood regions of the generated distribution, as shown by points $g_1$ and $g_2$. In scenarios where the real and generated distributions are closely aligned (top figure), the likelihood of generated images with respect to the real distribution, denoted as $L_g$, will be high, nearly matching the likelihood of real images, $L_r$, reflecting strong distributional similarity. When averaging over all generated images, the average likelihood is correspondingly higher. Conversely, when the real and generated distributions are dissimilar or more distant (bottom figure), the likelihood of generated images $L_g$ relative to the real distribution significantly decreases. As we average over all generated images in this case, the result is a notably lower overall likelihood compared to the case of aligned distributions, illustrating the increased distance between them and its impact on likelihood evaluation. Therefore, the likelihood of generated images with respect to the real data distribution serves as an effective metric for assessing the distance between two data distributions.

$$\mathbf{x}_p = p(\mathbf{x}_b) \quad \mathbf{x}_b \in \mathbb{R}^{H \times W \times C}, \mathbf{x}_p \in \mathbb{R}^{H/2 \times W/2 \times C} \tag{2}$$

$$\mathbf{x}_f \longleftarrow \mathbf{x}_p \quad \mathbf{x}_f \in \mathbb{R}^D, \quad \mathbf{x}_p \in \mathbb{R}^{H/2 \times W/2 \times C}, \quad D = H/2 \times W/2 \times C \tag{3}$$

$$\mathcal{L}_r = N(\mathbf{x}_f, \theta) \quad \mathcal{L}_r \in \mathbb{R}, \mathbf{x}_f \in \mathbb{R}^D \tag{4}$$

We first train the flow model $F(\theta)$ on the set of real images, $\mathcal{R}$ as shown in Eq. 5. This forces the flow model to give high likelihood for real images.

$$\mathcal{L}_r = F(\mathbf{x}_{in}, \theta) \quad \mathbf{x}_{in} \in \mathcal{R} \tag{5}$$

During the evaluation stage, we compute the average log-likelihood of all images in the real set, $\mathcal{R}$, using $F(\theta)$, and similarly evaluate the average log-likelihood of all images in the generated set, $\mathcal{G}$, with $F(\theta)$.

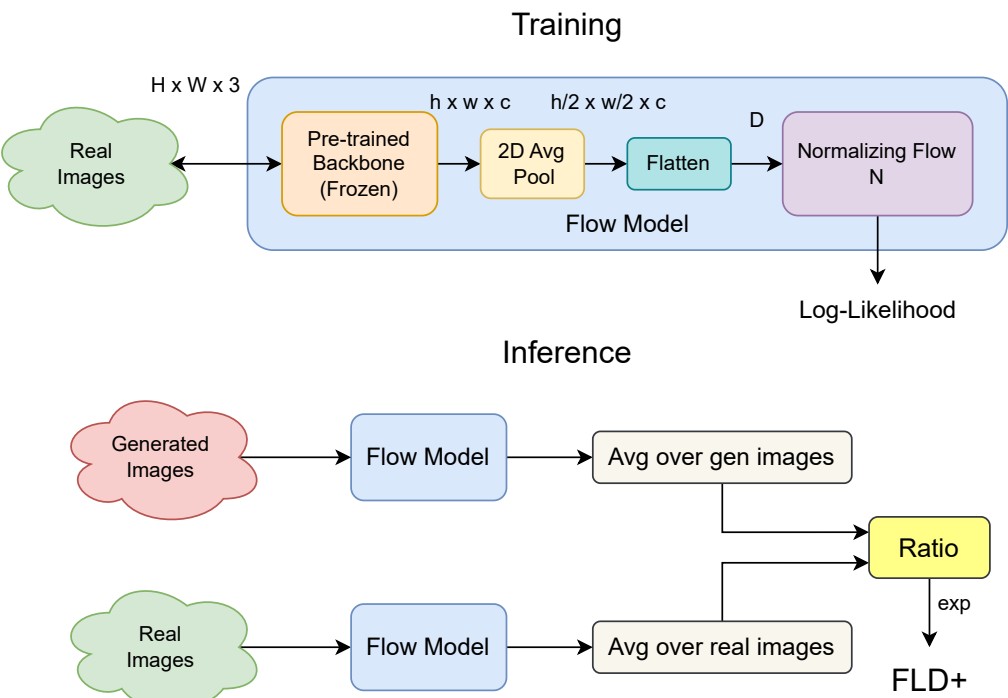

Figure 2: To compute FLD+, we start by training the flow model on real images. In the training phase, real images are processed through a frozen pre-trained vision backbone, where activations from the penultimate layer undergo average pooling and are then flattened before being passed to the normalizing flow. During the evaluation phase, both real and generated images are input into the flow model. Their log-likelihoods are calculated, averaged separately, and then the ratio of these averages is computed. Finally, this ratio is exponentiated to obtain the FLD+ metric.

We calculate the ratio of the average log-likelihood of the generated samples to that of the real samples, as illustrated in Fig. 2. This approach reflects the typically lower (more negative) log-likelihood values for generated samples compared to real samples. Finally, we take the exponential of the ratio to obtain the final FLD+ score (Eq. 6), representing the distance between the two distributions.

$$\text{FLD+} = \exp \left( \frac{\frac{\sum_{\mathbf{x} \in \mathcal{G}} \mathcal{L}_r(\mathbf{x})}{|\mathcal{G}|}}{\frac{\sum_{\mathbf{x} \in \mathcal{R}} \mathcal{L}_r(\mathbf{x})}{|\mathcal{R}|}} \right) \tag{6}$$

Since the normalizing flow $N$ within the flow model $F$ is trained exclusively on real images, it learns to compute likelihoods based on the real data distribution. As a result, when evaluating the likelihood for real data, $N$ yields higher likelihood values. If the generated data distribution closely aligns with the real data distribution, the average likelihood of images from the generated distribution will also be high and similar to those of the real data. Conversely, if the generated distribution deviates significantly from the real data distribution, the likelihood of generated images (as measured by $N$ with respect to the real distribution) will be much lower, indicating a larger distance between the two distributions.

### 3.3 Hypothesized Advantages of FLD+

**Stability with fewer evaluation images:** Because normalizing flows approximate the exact log-likelihood of each sample, which is a scalar quantity, it will likely produce a stable metric when averaged

over just a few hundred images. Other metric need tens of thousands of images to first estimate the assumed parametric distribution.

**Sensitivity to small image degradations:** Because normalizing flows are not based on an assumption of normality, they are better at modeling intricate data distributions and thus more sensitive and directionally correct at detecting image degradations of various kinds.

**Adaptability to new domains:** While it may seem paradoxical that FLD+ uses a feature extractor pre-trained on ImageNet for transfer learning while claiming adaptability to new domains, the use of a light-weight trainable normalizing flow module resolves this paradox. To address dataset bias from using only pre-trained CNN features, we introduce a normalizing flow after obtaining feature embeddings from the ImageNet pre-trained model. This normalizing flow is trained on domain-specific features even when computed using an out-of-domain extractor, effectively adapting the embeddings to the target domain. Additionally, because we are operating on lower-dimensional embeddings compared to the input data, the normalizing flow model of FLD+ is both data- and compute-efficient for retraining on target domains.

## 4 Experiments and Results

### 4.1 Datasets and Implementation Details

We used CelebA-HQ (Karras et al., 2018) dataset for our experiments to analyze the behaviour of the proposed metric and compare it with FID. We used CelebA-HQ at $256 \times 256$ resolution. All experiments were run on Nvidia A6000 GPU.

We used a rational-quadratic neural spline flow (Durkan et al., 2019) as the normalizing flow and use ImageNet pre-trained ResNet-18 (He et al., 2015) (with the final layer removed) as our backbone in the flow model for calculating FLD+. The feature tensors of shape $8 \times 8 \times 512$ was pooled using 2D average pooling to $4 \times 4 \times 512$ and then flatted as 8192 dimensional vector which is fed to the neural spline flow. Details of how different noise were added to images is provided in Appendix.

### 4.2 Monotonicity with Image Distortions

We evaluated whether FLD+ increases monotonically with rising levels of various image distortions and observed robust, desirable trends. Using 2,000 randomly selected images from the CelebA-HQ dataset, we applied various distortions, including Gaussian noise, Gaussian blur, and salt-and-pepper noise.

Previous studies (Jeevan et al., 2024; Jayasumana et al., 2023) have shown that FID often lacks a monotonic response and can inaccurately rate noisier images as higher quality than those with less noise, particularly when small amounts of Gaussian noise are introduced. In contrast, our results demonstrate that FLD+ maintains a consistent monotonic trend with increasing levels of Gaussian noise, as illustrated in Fig. 3.

Figure 10 and Fig. 5 illustrate the behavior of FLD+ when Gaussian blur and salt-and-pepper noise are applied to CelebA-HQ images. We observe that as distortion levels increase, the metric value consistently rises in a strict monotonic pattern, demonstrating the robustness of FLD+ as an evaluation metric.

### 4.3 Progressive Image Generation

Modern image generation models, such as diffusion models, iteratively refine noisy images in steps to produce high-quality outputs. Previous studies have shown that FID does not behave monotonically across denoising iterations (Jayasumana et al., 2023), making it unsuitable for evaluating these models, particularly in the final steps when noise levels are minimal. Figure 6 present FLD+ values for the final 100 iterations in a 1000-step diffusion process (Ho et al., 2020), demonstrating that FLD+ accurately and monotonically captures differences in image quality, even at low noise levels.

To assess the performance of our evaluation metric on state-of-the-art text-to-image diffusion models, which generate images based on text prompts, we utilized the Stable Diffusion-100K (Turley, 2023) dataset, selecting 1,000 prompts for our analysis. To evaluate whether our metric can accurately distinguish subtle

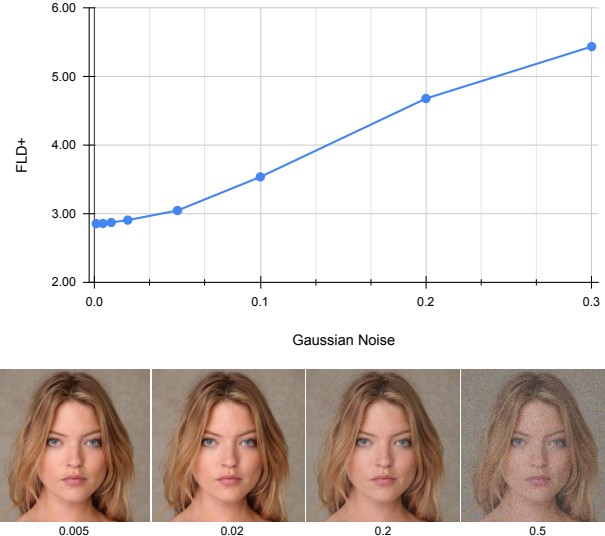

Figure 3: Monotonicity and robustness of FLD+ with increasing levels of Gaussian noise applied to images are shown (top). Images with progressively higher levels of Gaussian noise, displayed from left to right, are shown below with corresponding noise values, $\alpha$, indicated beneath each image (bottom).

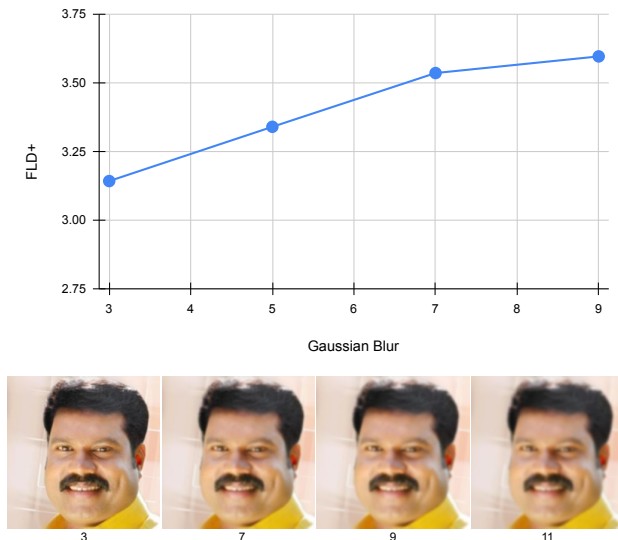

Figure 4: Monotonicity and robustness of FLD+ with increasing levels of Gaussian blur applied to images are shown (top). Images with progressively higher levels of Gaussian blur, displayed from left to right, have corresponding values of kernel size used for creating the blur, indicated below each image (bottom).

quality differences between images generated by a larger diffusion model (3.5 B parameter Stable Diffusion XL (Podell et al., 2023)) and those from a smaller (860 M parameter Stable Diffusion v1.5 (Rombach et al., 2022)), we calculated the FLD+ score by comparing the output images from these text-to-image diffusion models with real images in the Stable Diffusion-100K dataset (Turley, 2023). Our results in Table 1 demonstrate that the FLD+ metric reliably identifies images from the larger diffusion model as having higher quality compared to those generated by the smaller diffusion model. Results of similar evaluation of images generated by StyleGAN2 (Karras et al., 2020) is shown in Appendix.

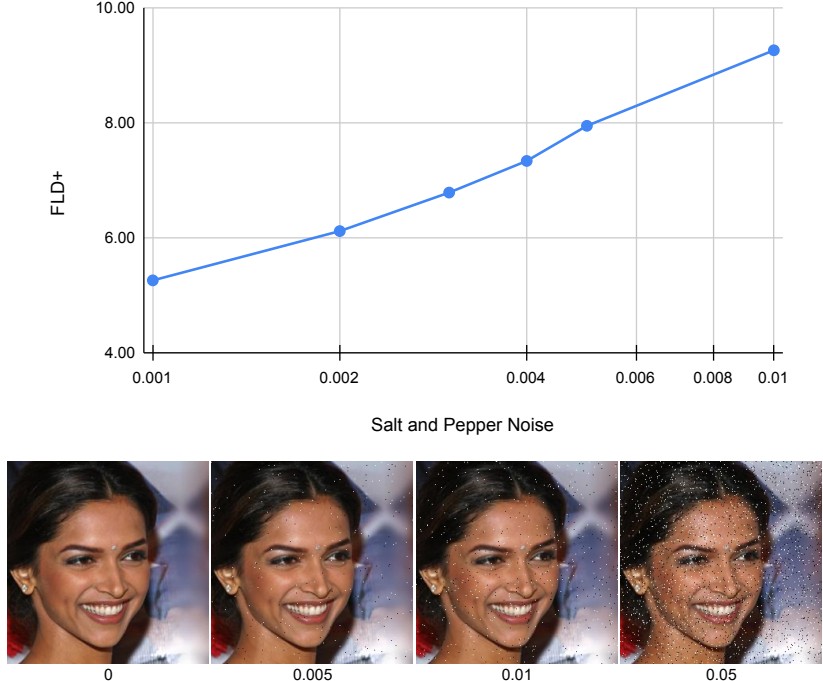

Figure 5: Monotonicity and robustness of FLD+ with increasing levels of salt-and-pepper noise applied to images are shown (top). Images with progressively higher levels of salt-and-pepper noise, displayed from left to right, have corresponding probability values that controls the proportion of pixels that will be corrupted, indicated below each image (bottom).

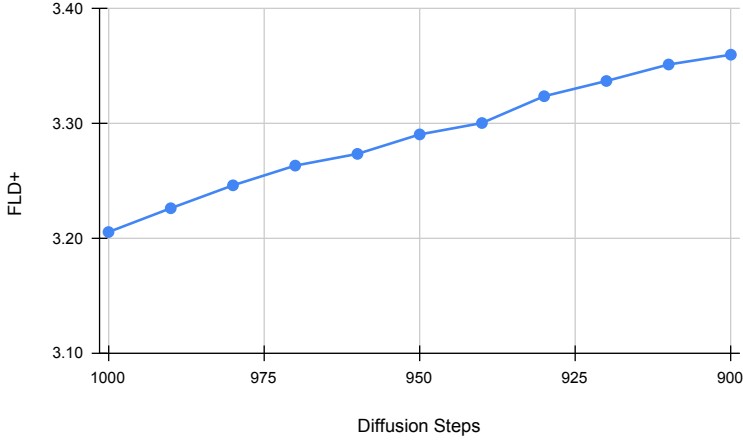

Figure 6: The graph illustrates the monotonicity of FLD+ when evaluating noise levels across different diffusion steps. As the number of diffusion steps increases, the noise in the images decreases, and FLD+ accurately reflects this reduction, demonstrating its ability to track changes in image quality throughout the diffusion process.

Table 1: When comparing the performance of text-to-image diffusion models, FLD+ effectively distinguishes the images generated by the larger Stable Diffusion XL model as higher quality than those produced by the smaller Stable Diffusion V1.5 model (mean ± std error).

| MODEL | # PARAM. | FLD+ |
|---|---|---|
| Stable Diffusion v1.5 (Rombach et al., 2022) | 0.9 B | $3.3277 \pm 0.0005$ |
| Stable Diffusion XL (Podell et al., 2023) | 3.5 B | $3.2167 \pm 0.0002$ |

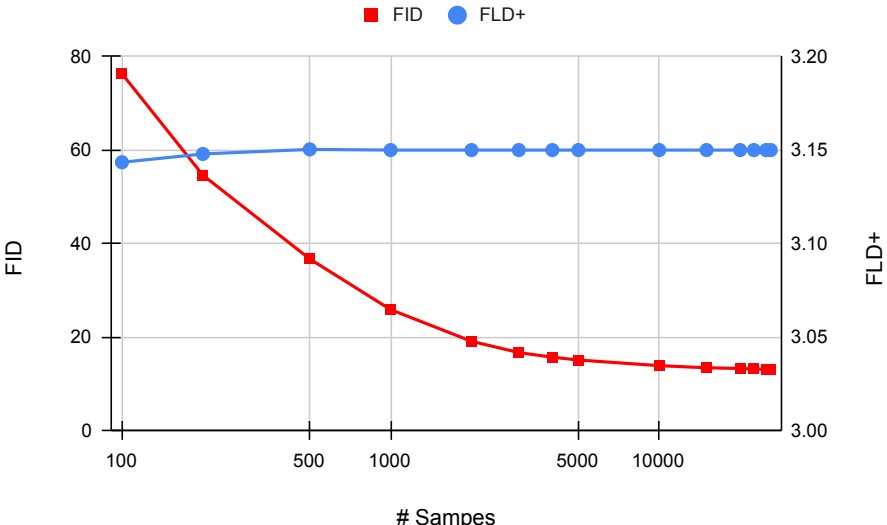

Figure 7: The mean values of FLD+ and FID across varying numbers of generated images clearly highlight FLD+'s superior sample efficiency, as it achieves reliable results with fewer than 200 samples. In contrast, FID requires more than 20,000 samples to produce a stable mean score. Standard deviations are provided in the Appendix.

## 4.4 Sample Efficiency

A major drawback of previous metrics, such as FID, is their lack of sample efficiency. These metrics require a large number of images—often in the tens of thousands—to produce a reliable estimate, necessitating a substantial amount of both real and generated data to accurately evaluate generative models. However, in data-scarce domains, FID becomes an unreliable measure of quality due to the limited availability of real data for computation. Even more recent methods, such as CMMD (Jayasumana et al., 2023), still require thousands of images, posing challenges for accurate evaluation in fields where data is limited.

Our FLD+ metric demonstrates high sample efficiency, achieving a stable value with just a few hundred samples. As illustrated in the Fig. 7, while FID requires over 20,000 images to reach stability, the FLD+ metric achieves similar stability with only a few hundred images (less than 300). This efficiency makes FLD+ an ideal evaluation metric in data-constrained scenarios, and it can even be used within the training loop during validation to assess model performance in real-time, without the need to wait for all epochs to complete before evaluating on the final dataset.

Since both FLD (Jeevan et al., 2024) and FLD+ require training the flow model on real data, we compared their training speeds using 30,000 images of size $256 \times 256$. While FLD takes 545 seconds per epoch, FLD+—with its smaller latent space—requires only 43 seconds per epoch, achieving a $12\times$ reduction in training time.

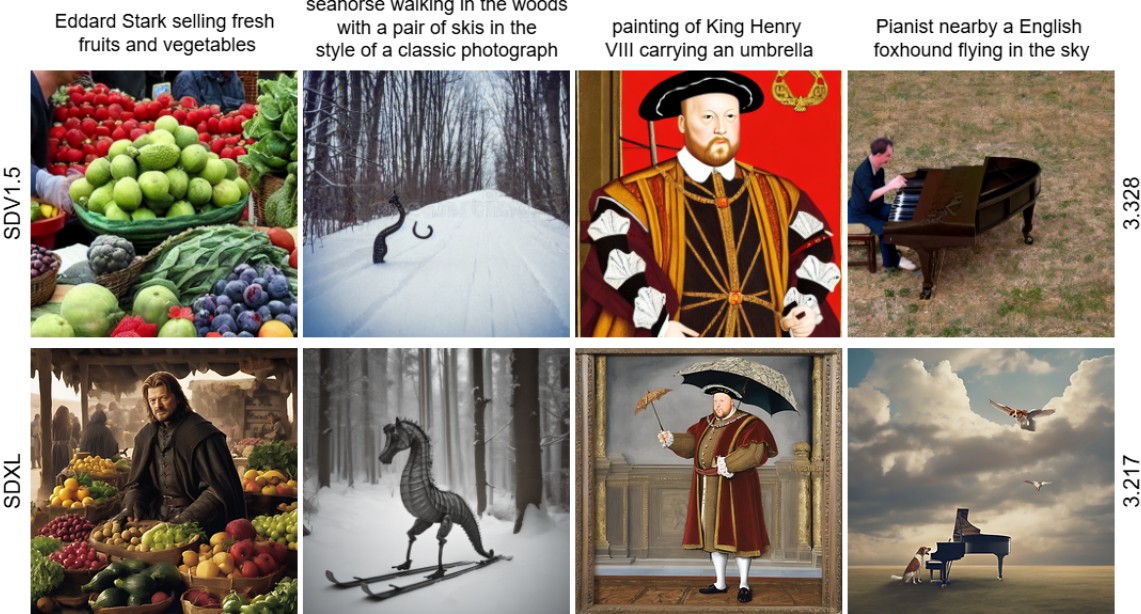

Figure 8: The behavior of FLD+ in comparing two text-to-image diffusion models reveals that FLD+ scores are lower for images generated by the larger Stable Diffusion XL model (third row, FLD+:3.217) indicating superior image quality from this model. In contrast, the smaller Stable Diffusion V1.5 model produces images with higher FLD+ scores (second row, FLD+: 3.328) reflecting their comparatively lower quality. The prompts used for generating the images are shown in first row.

We also compared the evaluation times required for reliably computing FID, FLD, and FLD+. FLD+ requires only 0.54 seconds to evaluate 200 samples, making it 6× faster than FLD, which takes 3.45 seconds. In comparison, FID takes 6.30 seconds to evaluate 200 samples but provides an unreliable estimate at this sample size. For a reliable FID evaluation, 20,000 images are needed, which takes 165.54 seconds, making FLD+ approximately 300× faster than FID.

## 5 Ablation Studies

**Backbones**: In addition to the ResNet-18 backbone used in our flow model (37 M parameters), we experimented with a smaller, low-parameter architecture by selecting the MobileNetV3-Small (Howard et al., 2019) model to assess whether our flow model (8 M parameters) would maintain monotonic behavior with increasing noise levels. Our results demonstrate that, even with the MobileNetV3-Small backbone, the FLD+ metric remains monotonic in response to multiple types of noise.

**Pooling Operations**: In addition to the average pooling used in our flow model, we experimented with max pooling to determine if it would also yield monotonic behavior. While we observed that max pooling does produce a monotonic response to increasing noise, max pooling was much less sensitive to variations in noise compared to average pooling. As a result, subtle differences in noise are not captured as effectively with max pooling, highlighting average pooling's superior sensitivity in detecting fine-grained noise variations.

## 6 Conclusions

In this work, we addressed the limitations of conventional evaluation metrics, such as FID, for evaluating generative models. Previous studies have shown that FID is not only unreliable but also data-intensive, requiring tens of thousands of images to provide a stable estimate. This dependency limits its efficiency and applicability, particularly in real-time or data-scarce scenarios. Furthermore, FID lacks strictly monotonic

behavior when evaluating images with increasing distortions, making it less effective in accurately reflecting changes in image quality.

To overcome these challenges, we developed an enhanced normalizing flow-based evaluation metric, FLD+. By leveraging an ImageNet pre-trained backbone and pooling, our approach effectively reduces image resolution and improves efficiency by working on a smaller latent space for computing likelihood. This architecture significantly improves computational efficiency, allowing for faster and more accurate evaluation.

Our experiments demonstrate that our FLD+ metric is robust, reliable and highly data-efficient, achieving stable estimates with fewer than 300 images. This makes it a practical choice for evaluating generative models in scenarios with limited data availability and for use within training loops to monitor performance in real-time.

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

## A  Details of Distortions

**Gaussian Noise:** We construct a noise matrix $N$ with values drawn from a $\mathcal{N}(0,1)$ Gaussian distribution and scaled to the range $[0,255]$. The noisy image is then computed by combining the original image matrix $X$ with the noise matrix $N$ as $(1-\alpha) \cdot X + \alpha \cdot N$, where $\alpha \in \{0, 0.001, 0.005, 0.01, etc\}$ determines the amount of noise added to the image. A larger value of $\alpha$ introduces more noise and the noisy image is clipped to ensure that all pixel values remain within the valid range $[0,255]$.

**Gaussian Blur:** The image is convolved with a Gaussian kernel with kernel size $k$. The kernel size controls the intensity of the blur, with larger kernels producing greater smoothing effects. The standard deviation ($\sigma$ is set to 0, allowing it to be automatically computed based on the kernel size. This approach simulates different levels of image degradation, ranging from minimal to significant blurring, by convolving each image with a Gaussian kernel. In this case, $k \in \{3, 5, 7, 9, 11\}$, and the Gaussian blur is applied uniformly across the image.

**Salt and Pepper Noise Addition:** To add salt and pepper noise to an image, random values are generated for each pixel. The probability $p$ controls the proportion of pixels that will be altered. For salt noise, pixels with random value less than $\frac{p}{2}$ are set to the maximum intensity value (255). Similarly, for pepper noise, pixels with random value greater than $1 - \frac{p}{2}$ are set to the minimum intensity value (0). The amount of noise is directly proportional to the value of $p$, here $p \in \{0, 0.001, 0.005, 0.01, etc\}$ with larger values of $p$ resulting in a higher number of noisy pixels in the image.

For experiments involving sampling efficiency, we used a CelebA-HQ pre-trained diffusion model for generating images (Chen, 2023). For experiments on checking the performance of our metric on progressive image generation, we used the original DDPM model (Ho et al., 2020).

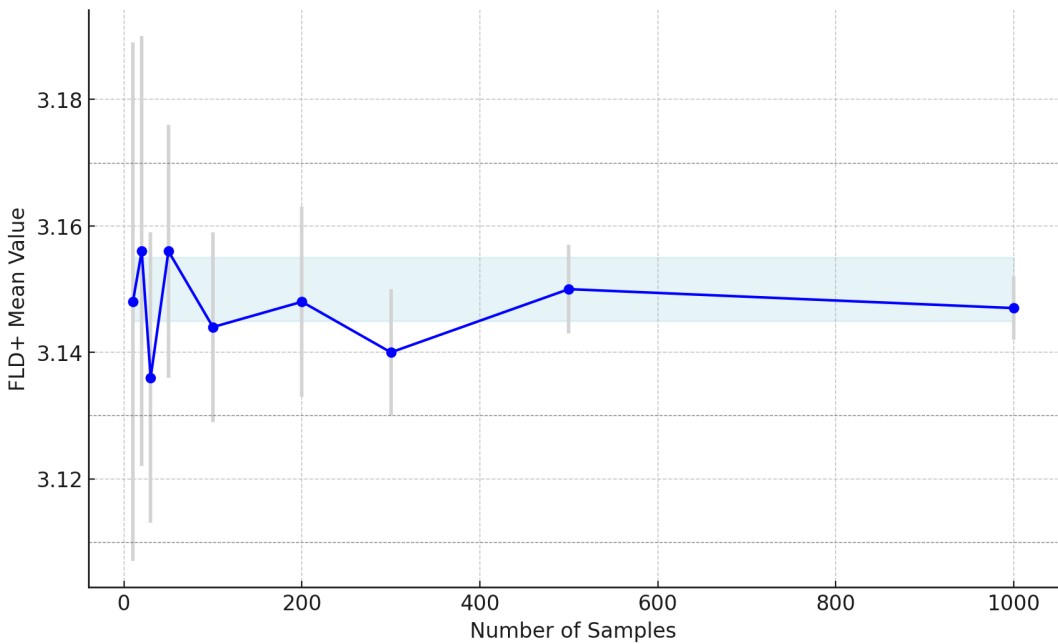

Figure 9: The behavior of FLD+ across different sample sizes clearly demonstrates that FLD+ achieves reliable results with fewer than 500 samples. The variance in FLD+ computed clearly goes down when sample size is more than 500. The mean FLD+ (blue) is reported after 10 runs for each sample size.

Table 2: FD, unbiased FD, and FLD+ values when the normality assumption is violated show the advantage of using FLD+. The leftmost image represents a reference mixture-of-Gaussian distribution, progressively from left to right the subsequent distributions deviate further from the true distribution keeping mean and standard deviation same. The FD of all the subsequent mixture distributions to the reference mixture distribution, calculated under the normality assumption, remain misleadingly zero (Jayasumana et al., 2023). In contrast, FLD+ accurately captures the increasing deviation from the reference distribution, demonstrating its robustness in scenarios where the normality assumption is violated.

| | | | | | | |
|---|---|---|---|---|---|---|
| FD | 0 | 0 | 0 | 0 | 0 | 0 |
| Unbiased FD | 0 | 0 | 0 | 0 | 0 | 0 |
| **FLD+** | 2.72 | 12.68 | 21.11 | 31.19 | 47.94 | 80.64 |

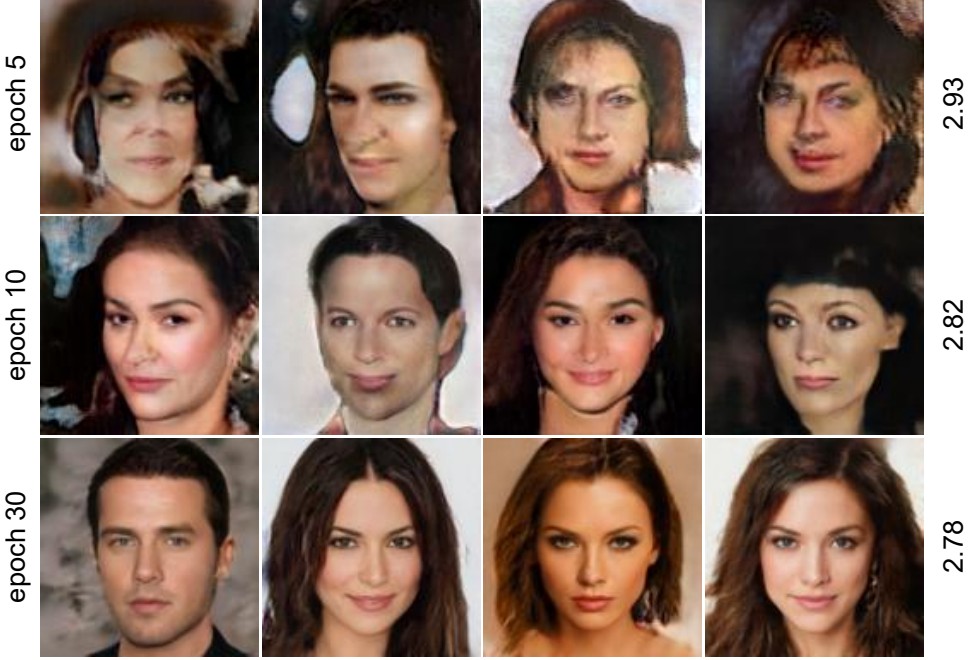

Figure 10: The figure illustrates the monotonicity of FLD+ when evaluating generated images across different training epochs for StyleGAN2 (Karras et al., 2020). As the number of training epochs increases, the distortion in the images decreases, and FLD+ accurately reflects this reduction, demonstrating its ability to track changes in image quality across the adversarial training epochs.

