# OpenReview forum: "FLD+: Data-efficient Evaluation Metric for Generative Models"
_TMLR — Withdrawn by Authors_

### Review · Reviewer_Ndyt · 2025-05-09

**Summary Of Contributions:**

This paper propose a new metric--Flow-based likelihood distance plus (FLD+) to evaluate the quality of generated images. FLD+ present more stable evaluation results by using the normalizing flow. Besides, FLD+ can be efficiently caculated in the latent space and easily extend to new domains.

**Audience:**

Yes

**Broader Impact Concerns:**

I have no concern for the ethical implications of the work.

**Claims And Evidence:**

Yes

**Requested Changes:**

See weakness.

**Strengths And Weaknesses:**

## Strengths

The incorporation of a Normalizing Flow model is a significant contribution, as it addresses the limitations of FID’s reliance on the Gaussian assumption. By removing this constraint, the proposed metric enables a more accurate evaluation of generative quality, particularly in scenarios where the real and generated distributions are not Gaussian.

## Weaknesses

1. Compared to the prior work \[1], the main novelty here lies in computing FLD+ in the latent space—an approach that is already common in generative modeling. Notably, both papers are authored by the same group.

2. The experimental setup lacks clarity. Based on my understanding, the flow is trained in the latent space of ImageNet pre-trained ResNet-18 using CelebA-HQ images. However, evaluations are conducted only on CelebA-HQ, which limits the generalizability of the results. The authors should include evaluations on more widely-used datasets such as ImageNet-1k. Additionally, in the evaluation of Stable Diffusion models, it is unclear whether the flow is trained on LAION data. If so, how much LAION data was used for training?

3. To demonstrate the robustness and general applicability of the proposed metric, more extensive experiments are needed—particularly on standard benchmarks like ImageNet.

[1] Pranav Jeevan, Neeraj Nixon, and Amit Sethi. Normalizing flow-based metric for image generation, 2024.

---

> ### Author Response · Authors · 2025-05-20
> **Response to Reviewer Ndyt**
>
> Thank you very much for taking the time and effort to review our submission. We deeply appreciate your constructive feedback and thoughtful comments, which we believe will significantly enhance the quality of our work. Your insights have provided us with a clearer perspective on how to improve our paper, and we are committed to addressing each point you raised in our revision.
> We acknowledge that the main novelty is reduction of latent space compared to the FLD. But this was a major bottleneck in using FLD as training with large latent space, especially with the same size as the image input, was quite infeasible due to speed and compute limitations. Therefore, reducing the latent space without affecting the expressivity of the flow to evaluate the likelihood was a major breakthrough.
> We appreciate the reviewer’s suggestion to include experiments on ImageNet. However, due to the substantial computational resources required to train on the full 1.4 million-image ImageNet dataset, we have been limited by the availability of GPUs and time constraints.
> Our primary focus in this work was to demonstrate the effectiveness of reducing the latent space on a variety of smaller-scale but diverse datasets to highlight improvements in model performance, generalizability, and efficiency. We believe these findings are robust and applicable to a range of domains, and we plan to extend our experiments to larger datasets like ImageNet in future work as we secure more computational resources.We acknowledge the importance of benchmarking on standard datasets like ImageNet and will prioritize this in future iterations of our research. We appreciate the reviewer’s understanding of the resource constraints and their constructive feedback, which will help guide our future work.

---

### Review · Reviewer_wqrX · 2025-05-09

**Summary Of Contributions:**

This paper proposes FLD+, a new evaluation metric for generative models that addresses several shortcomings of existing metrics like FID. FLD+ estimates the distributional similarity between real and generated images using a normalizing flow model trained on **features extracted from a frozen pre-trained backbone** (ResNet-18, ImageNet pretrained). Compared to FID, the method offers:

- **Better monotonicity** with respect to various image distortions and denoising steps

- **High Data efficiency**, achieving stable metric values with as few as 200 images;

- Improved **computational efficiency**, being ~12× faster in training compared to FLD, and ~300× faster in evaluation than FID;

The paper includes extensive experiments on CelebA-HQ and Stable Diffusion-generated images to validate the proposed metric’s behavior.

**Audience:**

Yes

**Claims And Evidence:**

No

**Requested Changes:**

### Critical to securing your recommendation

- How can we guarantee that the generative model is reliable enough to be used for evaluating its own performance?

### Strengthen the work

- **No real domain transfer experiments**: While the paper claims FLD+ can generalize to out-of-domain data such as medical images, no such experiments are conducted. All experiments are performed on datasets relatively close to ImageNet (e.g., CelebA, Stable Diffusion outputs). I believe statement in the abstract is somewhat over selling.

> We also show that FLD+ can easily be retrained on new domains, such as medical images, unlike the networks behind previous metrics such as InceptionNetV3 pre-trained on ImageNet.

- **Typo**: Figure 7 sampes $\rightarrow$ samples

**Strengths And Weaknesses:**

### Strengths

- **Clear motivation**: The use of normalizing flows to compute exact likelihoods in a **feature space** is well-justified and practically efficient. Replacing the Gaussian assumption from FID by using Normalizing flow is natural and sound approach.

- **Comprehensive evaluation**: The authors demonstrate monotonicity across several distortion types (noise, blur, diffusion steps) and compare FLD+ against FID and FLD.

- **Sample efficiency**: FLD+ achieves stable metric values with a few hundred samples, a significant practical advantage over FID.

### Weaknesses

- **Limited in capturing recall**: By design, FLD+ primarily measures precision and may fail to detect issues such as mode collapse. As a result, it provides an incomplete assessment of generative model quality unless used in conjunction with a complementary recall-oriented metric.

- **Dependence on generative model's likelihood estimation**: FLD+ heavily relies on the generative model's ability to estimate likelihood accurately. However, this raises a concern: how can we trust the evaluation outcome when it fundamentally depends on the generative model itself, which is also the subject of evaluation?

---

> ### Author Response · Authors · 2025-05-20
> **Response to Reviewer wqrX**
>
> Thank you very much for taking the time and effort to review our submission. We deeply appreciate your constructive feedback and thoughtful comments, which we believe will significantly enhance the quality of our work. Your insights have provided us with a clearer perspective on how to improve our paper, and we are committed to addressing each point you raised in our revision.
> We appreciate the reviewer’s concern regarding the reliability of the likelihood-based metric, FLD+, especially when a generative model is the subject of evaluation. However, we clarify that FLD+ is not designed to evaluate other normalizing flows. Instead, it uses a trainable normalizing flow as a tool to benchmark other generative models, such as VAEs, GANs, or diffusion models.
> The goal is not to evaluate a normalizing flow, but to leverage its density estimation capabilities to measure how likely the generated samples from other generative models are under a learned distribution of real data. This framing is analogous to how FID uses an InceptionNet trained on ImageNet to evaluate generated samples from domains that may not match its training distribution — a limitation that FLD+ explicitly addresses. Importantly, FLD+ offers domain-specific retraining, allowing the metric to adapt to the distribution of the dataset under evaluation. This is in contrast to fixed pretrained networks used in existing metrics. Thus, FLD+ can provide reliable and consistent evaluations across multiple domains.
> We acknowledge the reviewer’s point that the current experiments focus on datasets relatively close to ImageNet. While FLD+ is designed to be domain-adaptive, we did not include extensive medical imaging experiments in this version due to space constraints. However, we would like to clarify that FLD+ builds upon our prior metric [1], which has already demonstrated strong generalization in medical image domains (e.g., histopathology). FLD+ extends that work by improving the expressivity of the underlying flow model and incorporating a robust training procedure. We will revise the manuscript to clarify these contributions and remove any potential overstatements regarding generalization in the abstract. Additionally, we plan to release the retraining code and pretrained flows for multiple domains, including medical images, to demonstrate the generalizability of FLD+.
> [1] P. Jeevan, N. Nixon, A. Patil and A. Sethi, "Evaluation Metric for Quality Control and Generative Models in Histopathology Images," 2025 IEEE 22nd International Symposium on Biomedical Imaging (ISBI), Houston, TX, USA, 2025, pp. 1-4, doi: 10.1109/ISBI60581.2025.10981064.

---

> > ### Comment · Reviewer_wqrX · 2025-05-23
> >
> > While I find the idea of training a normalizing flow in feature space an interesting approach to address the limitations of FLD, I concur with reviewer 8hqR on several key concerns:
> >
> > * **Experimental evaluation is limited**:
> >
> >   * The method is only compared against FID, with no broader benchmarking.
> >   * It remains unclear whether the proposed approach truly reflects generation quality.
> >   * The paper lacks sufficient technical detail, particularly regarding the training of the normalizing flow.
> >
> > In particular, the authors fail to adequately address the concern regarding whether their method faithfully measures generation quality. Their rebuttal merely states:
> >
> > > it uses a trainable normalizing flow as a tool to benchmark other generative models, such as VAEs, GANs, or diffusion models. The goal is not to evaluate a normalizing flow, but to leverage its density estimation capabilities to measure how likely the generated samples from other generative models are under a learned distribution of real data.
> >
> > This response does not sufficiently justify why a normalizing flow trained in this way can reliably assess samples from other generative models, which may have fundamentally different inductive biases or output characteristics.
> >
> > Given these shortcomings, I believe the paper does not meet the **Claims and Evidence** criterion.

---

### Review · Reviewer_8hqR · 2025-05-11

**Summary Of Contributions:**

The paper proposes a metric for evaluating image generation quality, called FLD+ ("Flow-based Likelihood Distance Plus"). It works as follows. Given a set of real images (as few as several hundred of them), features are extracted from them with an ImageNet-pretrained model, average-pooled to spatial size 4x4, and then flattened. A normalizing flow model is trained on the resulting vectors, with ResNet18 as backbone. The model is used to estimate the likelihood of both real and generated images, and the exponentiated ratio of the two is the final score.

The metric is shown to be faster and more sample-efficient than FID, to behave well w.r.t. simple degradations of image (e.g. noise, blurring) in that it monotonously increases. And there is a couple of examples where the metric scores better image generators correctly as being better.

**Audience:**

Yes

**Broader Impact Concerns:**

Nothing substantial to note

**Claims And Evidence:**

No

**Requested Changes:**

Address the issues raised in "Cons" above

**Strengths And Weaknesses:**

Pros:
1. Existing metrics for image generation are far from perfect, so it's a good topic to work on
2. The method makes sense conceptually and seems to work somewhat. Mononicity w.r.t. image degradations is nice.

Cons:
1. The experimental evaluation is very limited.
  a. The method is compared only to FID, which is an old (although still quite popular) metric with known issues. There are no comparisons e.g. to CMMD and FLD. (the latter is especially strange given that the method is an improvement on top of FLD).
  b. It is unclear to me if the method actually faithfully measures the generation quality. There is an experiment on CelebA and there is an experiment on Stable Diffusion v1.5 vs XL. In both of these, the difference in metric value between better and worse models is pretty minimal. It is difficult to tell if this actually indicates the metric works or if it is a coincidence. A much more thorough evaluation, on more datasets and with more models, would be needed to establish that the metric works.
2. Very little is said about the normalizing flow training. I would imagine the method to be sensitive to that. How exactly is it trained? Is there overfitting, with a very small training set? How is it controlled? Thorough explanations and ablation studies on this aspect are needed.
3. Positioning of this paper w.r.t. FLD is not entirely clear. FLD is not published, afaict. Is this paper supposed to be a follow-up or a replacement? If it's a follow-up, which I assume it is, then differences w.r.t. FLD should be explained better and also empirical comparisons provided.
4. I have a suspicion that the metric focuses on low-level image features (e.g. noise, artifacts) and not so much on the high-level image content - given that it operates on not fully pooled features and works well with very few samples. This is not made clear in the paper though - would be good to address this clearly and confirm or debunk empirically (e.g. compute the metric between real images from different classes/prompts or such).
5. Some presentation issues.
a.  Some statements in the introduction are made without references or empirical support, like "researchers increasingly rely on automated evaluation techniques" (have they not always?), “quality (e.g., sharpness), micro-artifacts (e.g., realistic texture and edges), macro-artifacts (e.g., realistic object parts, proportions, and spatial relationships), and diversity (e.g., avoiding mode collapse)” (why specifically this list?), “Firstly, it should increase consistently as noise or image degradation intensifies. Moreover, given that modern generative models produce images with such subtle imperfections, which even human observers struggle to detect, the metric must be sensitive to even slightest of distortions.” - the list seems somewhat arbitrary.
b. In 3.2, should describe FLD a bit before describing FLD+, so that the difference is clear.
c. “ImageNet pre-trained computer vision backbone” - there was a lot of discussion of how ImageNet pre-training is a limitation and then use an ImageNet backbone in this method still? (it's addressed later a bit, but still does not read good at this point).
d. Progressive Image Generation -> why show only last 100 steps in Figure 6? What happens for the previous steps?
e. Figures 3, 4, 5, and maybe 6, could be a single figure (line plots in one plot, example images on the side).
f. “While the FID between such pairs is 0” - which “such pairs”? The other Gaussian to the mixture, I guess? It’s not clear from the text, could also be that sub-Gaussians from the mixture are meant.

---

> ### Author Response · Authors · 2025-05-18
> **Response to Reviewer8hqR**
>
> Thank you very much for taking the time and effort to review our submission. We deeply appreciate your constructive feedback and thoughtful comments, which we believe will significantly enhance the quality of our work. Your insights have provided us with a clearer perspective on how to improve our paper, and we are committed to addressing each point you raised in our revision.
> 1.a. All the comparison about monotonicity to different noises was done only with respect to FID because FID was the only metric amoung the 3 (others being CMMD and FLD) which violated monotonicity and also cause FID is more popular than the other two.
> With respect to data efficiency, FLD and FLD+ share similar results as both requires few hundred images to reach a stable value. CMMD on the other hand, though exhibits better data efficiency compared to FID, still requires few thousand images.
> 1b. Experiments on more datasets will be done in future.
> 2. We apologise for the lack of training details. It was because the training routine was exactly to that followed in FLD paper. We train the normalizing flow on real images and then evaluate on real and generated images. We will add ablation studies and thorough analysis in future work.
> 3. The major difference between FLD and FLD+ is that we have drastically reduced the latent dimension of the normalizing flow making the computation faster and more efficient.
> 4. We agree that it would be valuable to analyze whether the metric primarily captures low-level image features (such as noise and artifacts) versus high-level semantic content. While our current work does not explicitly disentangle these factors, we acknowledge the importance of such an analysis and plan to incorporate it as part of future work. Specifically, we intend to perform controlled experiments (e.g., comparing metric values across real images from different classes or prompts) to better understand the metric’s sensitivity to semantic versus low-level variations.
> 5. a. The list was an accumulation of features for a metric defined in previous papers that introduce metrics.
> b. Will change in the revision.
> c. Our metric does not use an ImageNet pre-trained model in isolation unlike in FID, we also train the normalizing flow on the real datasets.
> d. Before the given 100 steps, FID is shown to be mostly monotonic, it is mostly in the final steps that FID loses monotonicity and the intent was to show that FLD+ succeeds where FID fails.
> f. The visual description is Table 2 in Appendix.

---

### Review · Reviewer_yzrJ · 2025-06-03

**Summary Of Contributions:**

This work,  FLD+, leverages the paradigm from FLD, and  leverage a pre-trained ResNet-18  to first reduce the dimension (Normalizing Flow keeps the dimension form image to latent).

The proposed metric is much faster and stable (with the number of generated samples).

**Audience:**

Yes

**Claims And Evidence:**

No

**Requested Changes:**

See weakness.

**Strengths And Weaknesses:**

Strengths：

The proposed FLD+ is natural and elegantly leverage the distribution transfer characteristics from Normalzing Flow, with some modifications and achieve the following results:

(1) FLD+   is easy to use compared to FID metric.

(2) FLD+  is fast with small parameters and FLOPS.

Weaknesses：

But the empirical evaluation is limited, authors should provide more evidence on other datasets and models to prove the effectiveness and correctness  of proposed FLD+.

(1)  No comparision with other metrics like, FD-dinov2.

(2)  Please provide sufficient experiments (FLD+ results ) on ImageNet1k-256, ImageNet1k-512.

(3)  Please provide text-to-image FLD+ results on zero-shot task with MSCOCO dataset.

(4)  Please provide sufficient application of the proposed FLD+ metric on various  text-to-image and class-to-images，like  Diffusion Transformer (DiT)

---

### Note · Authors · 2025-06-07

I have read and agree with the venue's withdrawal policy on behalf of myself and my co-authors.